# Longitudinal Chest X-ray Scores and their Relations with Clinical Variables and Outcomes in COVID-19 Patients

**DOI:** 10.3390/diagnostics13061107

**Published:** 2023-03-15

**Authors:** Beiyi Shen, Wei Hou, Zhao Jiang, Haifang Li, Adam J. Singer, Mahsa Hoshmand-Kochi, Almas Abbasi, Samantha Glass, Henry C. Thode, Jeffrey Levsky, Michael Lipton, Tim Q. Duong

**Affiliations:** 1Department of Radiology, Renaissance School of Medicine, Stony Brook University, Stony Brook, NY 11794, USA; 2Department of Family Medicine, Renaissance School of Medicine, Stony Brook University, Stony Brook, NY 11794, USA; 3Department of Emergency Medicine, Renaissance School of Medicine, Stony Brook University, Stony Brook, NY 11794, USA; 4Department of Radiology, Albert Einstein College of Medicine and Montefiore Medical Center, Bronx, NY 10461, USA

**Keywords:** coronavirus, chest radiograms, computed tomography, ground glass opacity, consolidation, infiltrates

## Abstract

**Background:** This study evaluated the temporal characteristics of lung chest X-ray (CXR) scores in COVID-19 patients during hospitalization and how they relate to other clinical variables and outcomes (alive or dead). **Methods:** This is a retrospective study of COVID-19 patients. CXR scores of disease severity were analyzed for: (i) survivors (*N* = 224) versus non-survivors (*N* = 28) in the general floor group, and (ii) survivors (*N* = 92) versus non-survivors (*N* = 56) in the invasive mechanical ventilation (IMV) group. Unpaired *t*-tests were used to compare survivors and non-survivors and between time points. Comparison across multiple time points used repeated measures ANOVA and corrected for multiple comparisons. **Results:** For general-floor patients, non-survivor CXR scores were significantly worse at admission compared to those of survivors (*p* < 0.05), and non-survivor CXR scores deteriorated at outcome (*p* < 0.05) whereas survivor CXR scores did not (*p* > 0.05). For IMV patients, survivor and non-survivor CXR scores were similar at intubation (*p* > 0.05), and both improved at outcome (*p* < 0.05), with survivor scores showing greater improvement (*p* < 0.05). Hospitalization and IMV duration were not different between groups (*p* > 0.05). CXR scores were significantly correlated with lactate dehydrogenase, respiratory rate, D-dimer, C-reactive protein, procalcitonin, ferritin, SpO2, and lymphocyte count (*p* < 0.05). **Conclusions:** Longitudinal CXR scores have the potential to provide prognosis, guide treatment, and monitor disease progression.

## 1. Introduction

Coronavirus disease 2019 (COVID-19) [1,2,3] has already infected 676 million people and killed more than 6.88 million worldwide (14 March 2023). The widespread outbreaks and recent spikes around the world, and the likelihood of recurrences have strained and will continue to strain healthcare resources. Radiological imaging of the lung is an essential tool in evaluating COVID-19 lung infection. In the early days of the pandemic, computed tomography (CT) [4] was used in China when reverse transcription polymerase chain reaction (RT-PCR) was less reliable and had a long turnaround time [5,6]. CT is, however, prone to cross-contamination and, thus, it is not widely used in the context of COVID-19 in the United States and elsewhere in the world, especially in the intensive care setting, due to the risk of cross-infection. By contrast, a portable chest X-ray (CXR) is convenient, readily available, can be brought to the patient’s bedside, and can be readily disinfected between uses [7,8,9,10,11,12,13,14,15,16]. Although a CXR has inferior diagnostic quality to CT, CXRs can be used to visualize characteristic ground-glass opacities and consolidation in the lungs associated with COVID-19 infection, helping with clinical diagnosis [17]. CXRs have become increasingly relevant in COVID-19 circumstance because a disproportionally large percentage of COVID-19 patients are put on invasive mechanical ventilators for a much longer duration compared with other similar lung infections [18]. Improved understanding of the temporal progression and disease severity of COVID-19 lung infection on CXRs has become more urgent.

A few recent studies have related initial CXR scores of COVID-19 patients presenting to the emergency department (ED) to clinical outcomes, such as mortality, escalated care, length of hospitalization, and duration on a ventilator [19,20,21,22,23]. Results remain inconsistent and controversial, with some reporting good correlation of CXR scores with these clinical outcomes while others did not. To our knowledge, there has been no systematic evaluation of the temporal characteristics of lung CXR scores in COVID-19 patients and how these characteristics can be judiciously used to inform clinical decision-making. As such, the potential of CXRs in the COVID-19 pandemic has not yet been fully realized.

This study sought to determine the prognostic values of *longitudinal* CXR scores in COVID-19 patients admitted only to the general floor (GF group) and patients treated with invasive mechanical ventilation (IMV group). For this, we analyzed CXR scores at different time points in those two groups using various statistical methods. We first demonstrated the demographics, comorbidities, vital signs, and laboratory values of those two groups, as these could be confounding factors. We then demonstrated our CXR scoring method. With those foundations, we then analyzed: (i) CXR scores of the general floor (GF) COVID-19 patients at the time of admission versus at outcome (discharged alive or dead) stratified by survivors and non-survivors; (ii) CXR scores of invasive mechanical ventilation (IMV) COVID-19 patients at the time of intubation versus at outcome stratified by survivors and non-survivors; (iii) CXR scores of IMV COVID-19 patients longitudinally for the first five days on IMV stratified by survivors and non-survivors; and (iv) the relationship between longitudinal CXR scores and clinical variables such as laboratory test results and vital signs. The results of these analyses were presented in the results section. We then discussed our results, mainly the temporal relationship between CXR scores and mortality as well as the relationship between CXR scores and clinical variables in the two groups in the discussion section. Limitations of the study and future directions were discussed as well.

## 2. Materials and Methods

Patient selection and inclusion criteria: This retrospective study was approved by the University Institutional Review Board with an exemption of informed consent. Our study followed the Strengthening of Reporting of Observational Studies in Epidemiology (STROBE) reporting guidelines for cross-sectional studies (http://www.equator-network.org/reporting-guidelines/strobe/, accessed on 14 February 2023). These hospital data have been used by others to address other clinical questions. There were 4542 persons under investigation (PUI) who presented to the ED at University Hospital between 8 March 2020 and 30 June 2020, of whom 1975 tested positive and were hospitalized. COVID-19 status was determined by RT-PCR. We further excluded patients with an incomplete history of comorbidities, no CXR within three days of their ED visit, and fewer than two CXRs. We should note that for regular floor patients, CXRs were typically acquired once, with repeats only as clinically indicated. For mechanically intubated patients, CXRs were usually performed essentially daily in our hospital (other hospital practices could differ), to check the position of the lines and tubes. The final sample sizes were: (1) survivors (*N* = 224) versus non-survivors (*N* = 28) for the GF group, and (2) survivors (*N* = 92) versus non-survivors (*N* = 56) for the IMV group. We should note that, essentially, all patients escalated to intensive care were placed on invasive mechanical ventilation in our cohort.

The primary outcome was mortality. The following data were obtained: (i) duration of hospitalization; duration on IMV; duration post-IMV in the hospital; (ii) CXR scores and clinical variables for the GF group stratified by survivors and non-survivors at the time of ED admission and at outcome (discharged alive or dead); (iii) CXR scores and clinical variables for the IMV group stratified by survivors and non-survivors at the time of intubation and at outcome; and (iv) CXRs scores and clinical variables of five consecutive days on IMV stratified by survivors and non-survivors.

CXR scores: A group of four board-certified chest radiologists of 10–20 years of experience and two radiology residents in training under attending supervision scored the CXRs for disease severity using the following criteria based on geographical extent and degree of opacity. The geographical extent score of 0–4 was assigned to each of the right and left lung fields depending on the extent of involvement with ground glass opacity or consolidation: 0 = no involvement; 1 = <25%; 2 = 25–50%; 3 = 51–75%; 4 = >75% involvement. The right and left lung were scored separately and were added together. The degree of opacity score of 0–4 was assigned to each of the right and left lungs as: 0 = no opacity; 1 = ground glass opacity; 2 = mix of consolidation and ground glass opacity (less than 50% consolidation); 3 = mix of consolidation and ground glass opacity (more than 50% consolidation); 4 = complete white-out. The right and left lung were scored separately and added together. In short, the geographical extent score ranged from 0–8, and the opacity score ranged from 0–8. Each CXR was independently scored by two raters who were blinded to the clinical data and the average of the scores by the two raters was calculated as the final score.

Clinical variables: Demographic information, chronic comorbidities, vital signs, and laboratory test results were collected. Demographic information included age, sex, ethnicity and race. Chronic comorbidities included smoking, diabetes, hypertension, asthma, chronic obstructive pulmonary disease, coronary artery disease, heart failure, cancer, immunosuppression and chronic kidney disease. Vital signs included heart rate (HR), respiratory rate (RR), pulse oxygen saturation (SpO2), systolic blood pressure (SBP), diastolic blood pressure (SBP) and temperature (temp). Laboratory test results included C-reactive protein [CRP], D-dimer (DD), ferritin, lactate dehydrogenase (LDH), lymphocytes (lymph), procalcitonin (procal), alanine aminotransferase (ALT), brain natriuretic peptide (BNP), creatinine (Cr), and troponin (TNT). These clinical variables were obtained for the GF group and IMV group stratified by survivors and non-survivors at the time of ED admission or initiation of IMV *and* at the time of outcome (discharged alive or dead).

Statistical analysis: Statistical analysis was performed with IBM SPSS software (v26, Armonk, NY, USA). Group comparison of categorical variables in frequencies and percentages used chi-squared or Fisher exact tests. Group comparison of continuous variables in medians and interquartile ranges (IQR) used the Mann–Whitney U test. Intraclass correlation coefficient [24] was calculated to assess inter-reader agreement of the CXR scores. Unpaired *t*-tests were used to compare survivors and non-survivors and time points. Comparison across multiple time points used two-way repeated measures ANOVA with the inclusion of group, day and group * day interaction as the independent variables. *p* values for post-hoc *t*-tests were adjusted with the Bonferroni–Holm correction for multiple comparisons. A *p* < 0.05 was considered statistically significant unless otherwise specified.

## 3. Results

### 3.1. Patient Characteristics

Descriptive statistics of demographics and comorbidities of the GF group and the IMV group were compared between survivors and non-survivors as in Table 1. In the GF group, mortality was significantly associated with age, ethnicity, race, hypertension, COPD, coronary artery disease, heart failure, and chronic kidney disease (*p* < 0.05). In the IMV group, mortality was only significantly associated with age, smoking, hypertension, COPD, coronary artery disease, and heart failure (*p* < 0.05).

Clinical variables which include vital signs and laboratory test results of the GF group at ED admission and at outcome by survivors and non-survivors were presented in Table 2A. At ED admission, essentially all these clinical variables (respiratory rate, SpO2, temperature, BNP, CRP, D-dimer, LDH, leukocyte, lymphocyte, procalcitonin) were significantly different between survivors and non-survivors (*p* < 0.05). At outcome, clinical variables (except lymphocytes, (*p* < 0.05)) were not significantly different between survivors and non-survivors (*p* > 0.05), which is due to the small sample size because these clinical variables were not generally obtained for GF patients prior to discharge. Similarly, the vital signs and laboratory test results of the IMV group at the time of intubation and at outcome by survivors and non-survivors are presented in Table 2B. At the time of intubation, none of the clinical variables (except D-dimer and leukocyte (*p* < 0.05)) was significantly different between survivors and non-survivors (*p* > 0.05). At outcome, essentially all these clinical variables (respiratory rate, SpO2, temperature, CRP, D-dimer, LDH, leukocytes, lymphocytes, procalcitonin) were significantly different between survivors and non-survivors (*p* < 0.05).

### 3.2. CXR Scores

Examples of CXRs with a different geographic extent and different opacity scores are demonstrated in Figure 1. CXRs of COVID-19 positive patients showed hazy opacities and/or airspace consolidation, with a predominance of bilateral, peripheral, and lower lung zone distribution. Each geographic score and opacity score ranged from 0 to 8, with a higher score indicating worse disease severity. Each CXR was rated independently by two raters. The inter-reader agreement of CXR scores assessed by intraclass correlation coefficient was 0.93 (95% CI: 0.93–0.94) for the geographic score, and 0.88 (95% CI: 0.86–0.89) for the opacity score, indicating excellent inter-rater agreement. The correlation (Pearson’s correlation coefficient = 0.69) between the extent score and opacity score was moderate.

CXR scores of the GF group obtained at ED admission and at outcome were stratified by non-survivors and survivors (Figure 2). Geographic and opacity scores behaved similarly, and they are discussed together, with geographic scores generally yielding bigger differences. Geographic and opacity scores were significantly higher (worse disease severity) in non-survivors compared to survivors at both time points (*p* < 0.05). Non-survivor scores significantly worsened at the second time point compared to the first time point (*p* < 0.05), but survivor scores did not (*p* > 0.05).

CXR scores of the IMV group at intubation and at outcome were stratified by non-survivors and survivors (Figure 3). Scores were not significantly different between non-survivors and survivors at the time of intubation (*p* > 0.05) but were significantly different at outcome (*p* < 0.05). Both survivors and non-survivors showed significant improvement in scores (*p* < 0.05), but survivors showed a bigger improvement (*p* < 0.05). Comparing the GF and IMV patients, the CXR scores of the GF survivors were lower than those of the IMV survivors by 1 to 4 points on average out of a maximum of 8 points, whereas the CXR scores of the GF and IMV non-survivors were similar.

CXR scores were plotted on five consecutive days on IMV (Figure 4). The scores were not significantly different between non-survivors and survivors on day 1 on IMV (*p* > 0.05) but diverged on subsequent days. Geographic scores were significantly different between groups at 2, 3, 4 and 5 days on IMV (*p* < 0.05). Opacity scores were significantly different between groups at 2 and 4 days on IMV (*p* < 0.05).

### 3.3. Histograms of Days in the Hospital and Duration on Ventilator

The durations of hospitalization of the general floor group were stratified by survivors (*N* = 224) and non-survivors (*N* = 28) (Figure 5A). The number of days of hospitalization of the general floor group was not significantly different between non-survivors (median 4.5 days [IQR:2, 9.5]) and survivors (median 5 days [IQR = 3, 7], *p* > 0.05). The histograms of duration of hospitalization of the IMV group stratified by survivors (*N* = 92) and non-survivors (*N* = 56) (Figure 5B). The number of days of hospitalization was not significantly different between non-survivors (median 12.5 days [IQR:6.5, 22]) and survivors (median = 18.5 days [IQR = 13, 26], *p* > 0.05). IMV patients were in the hospital markedly longer than GF patients (*p* < 0.01 for both survivors and non-survivors).

The duration on IMV and post IMV were stratified by survivors (*N* = 92) and non-survivors (*N* = 56) (Figure 6). The number of days on IMV was not significantly different between non-survivors (median 11 days [IQR:5, 19]) and survivors (median 10 days [IQR = 7, 19], *p* > 0.05). The number of days post-IMV was significantly different between non-survivors (median 0 days [IQR:0, 0]) and survivors (median 8 days [IQR:7, 14], *p* < 0.05).

### 3.4. Association of CXR with Laboratory Values and Outcomes

The associations between CXR scores with clinical variables were estimated using correlation analysis (Table 3). Both geographic and opacity CXR scores were significantly correlated with LDH, RR, D-dimer, CRP, procalcitonin, ferritin, SpO2, and lymphocyte count. The geographic score was correlated with WBC, and opacity score was correlated with troponin and HR. No CXR scores were significantly associated with SBP, temperature and BNP. The clinical variables with the highest correlation included LDH, RR, D-dimer and CRP.

The association between CXR scores and duration of hospitalization, IMV and post-IMV were estimated using a correlation stratified by survivals and non-survivors (Table 4). In the GF group, CXR scores were not correlated with the hospitalization duration (*p* > 0.05). Among the IMV non-survivors, CXR scores were correlated with hospitalization and IMV durations (*p* < 0.05), but not post-IMV duration. Among the IMV survivors, CXR scores were not correlated with hospitalization, IMV and post-IMV duration (*p* > 0.05) except for the geographic score with hospitalization.

## 4. Discussion

This study characterized the relationship between longitudinal CXR scores and clinical outcomes (mortality, hospitalization duration, IMV duration, and clinical variables) in the general floor patients and mechanically ventilated patients. The major findings are: (i) GF non-survivor CXR scores were significantly worse at admission relative to GF survivor scores, and GF non-survivor CXR scores worsened at outcome whereas GF survivor CXR scores did not; (ii) IMV non-survivor and survivor CXR scores were similar at intubation, while both improved at outcome, but survivor scores showed larger improvement; (iii) hospitalization or IMV duration were not significantly different between non-survivors and survivors; (iv) CXR scores were significantly correlated with LDH, RR, D-dimer, CRP, procalcitonin, ferritin, SpO2, and lymphocyte count; and (v) IMV non-survivor CXR scores were correlated with IMV duration, but GF CXR scores were not correlated with hospitalization duration.

### 4.1. Temporal Progression of CXR Scores

GF survivors showed significantly less severe lung involvement relative to GF non-survivors, IMV survivors, and IMV non-survivors at admission. These are not unexpected, but indicate correlation of severity of COVID-19 infection with likelihood of mortality, and CXRs could provide clinically useful information in a quantitative manner. Interestingly, GF non-survivors showed similar lung involvement severity compared to IMV patients at admission. This is not unexpected because most of the GF non-survivors were placed on comfort care; they would have been upgraded to IMV if they were full code. It is worth noting that the mortality outcome could thus depend on the patients’ will as well as the treatments they received.

GF survivor lung involvement did not improve or worsen from admission to outcome, while GF non-survivor lung involvement worsened. It is surprising that GF survivor lung involvement did not improve. A possible reason is that GF survivor CXR scores were low to begin with and the CXR abnormality did not fully resolve at the time of discharge. It is also possible that GF survivors were hospitalized for shorter durations compared to GF non-survivors, and it may take time for lung abnormality to resolve. Nonetheless, these findings suggest that it is not necessary for CXR abnormalities to be completely resolved prior to hospital discharge.

By contrast, IMV non-survivors and survivors showed similarly severe lung abnormality at IMV admission. The severity of lung infection in both IMV non-survivors and survivors was worse than that of GF survivors but was similar to that of GF non-survivors. Both IMV non-survivors and survivors showed improvement in CXR scores over five days on IMV and at discharge, with IMV survivors showing a larger improvement. Taken together, these observations suggest that an improvement in CXR scores is associated with IMV treatment. Other treatments under escalated care could also play a role in improving CXR scores.

While there are multiple publications on non-longitudinal chest X-ray scores in COVID-19 patients [19,20,21,22,23], there have been no studies on longitudinal chest X-ray scores. We believe our comparisons of radiologist CXR scores at admission, at pre-IMV, and at IMV longitudinally between survivors and non-survivors are novel. Our results show that CXR scores at admission and at intubation differed from those at outcome, supporting the notion longitudinally that CXR scores could inform the clinical outcome and guide clinical care better than non-longitudinal chest X-ray scores.

Longitudinal CXR scores offer important insights into disease progression in COVID-19 patients and facilitate disease management. This includes offering patients more supportive measures (such as oxygen supplementation and medications) and escalated care. In the context of ICU, CXR and CXR scores may be used to decide when to intubate, extubate, and reintubate as well as administer treatment regimens.

### 4.2. Geographic versus Opacity Scores

The trends for geographic and opacity scores are simlar overall. However, the geographic scores appeared to show large differences, i.e., were likely more sensitive in lung disease severity. The geographic scores reflect the extent of lung involvement, whereas the opacity scores reflect the degree of opacity. These findings suggest that the extent of lung involvement is more informative than the degree of opacity the lung appears. These observations suggest that clinicians and radiologists could pay more attention to the extent of lung involvement instead of the degree of involvement when assessing CXRs of COVID-19 patients.

Radiological scoring is widely used to stage lung disease severity, usually based on CT but not on CXRs because CT offers better sensitivity [25]. CT is not used in COVID-19 circumstances in most parts of the world because the equipment and suite are more difficult to disinfect and thus create concerns about cross-contamination of equipment, medical staff and patients. Quantitative CXR scoring is generally not a common practice in radiology. Radiologist reports of CXRs (including those of COVID-19) are qualitative. Our scoring system was adapted from those by Warren et al. [26] and Wong et al. [27]. In establishing our severity scoring system, a group of six chest radiologists worked together to reach consensus by evaluating two dozen images of portable CXRs of COVID-19 patients. In our scoring approach, the right and left lung were scored separately and added together. We additionally explored the sum and product of the geographic and opacity scores, and the results were similar. A few similar radiographic scoring systems have been used on COVID-19 CXRs in other studies [19,20,21,22,23], and most were based on a scale of 0–3. Each scoring system has its advantages and disadvantages. A simpler scoring system is likely to be easier to use and is efficient but may not have adequate dynamic range. A more sophisticated scoring system is likely to capture more information but may be more difficult to use and take a longer time to score.

### 4.3. CXRs Correlation with Other Clinical Variables

For the GF group, essentially all the tabulated clinical variables were significantly different between survivors and non-survivors at ED admission, but none of the tabulated clinical variables was significantly different between survivors and non-survivors. This was because of the small sample size because these clinical variables were not generally obtained for GF patients prior to discharge as doing so was unnecessary. The differences were qualitatively similar to the differences at ED admission overall. For the IMV group, essentially none of the tabulated clinical variables was significantly different between survivors and non-survivors at the time of intubation but they were significantly different at outcome. These findings are not unexpected. Both survivors and non-survivors had similar CXR disease severity prior to mechanical ventilation treatment, but survivors showed improvement in CXR severity. These findings suggest that mechanical ventilation treatment improved CXR scores and patient overall outcomes. They also suggest that CXR scores are informative and could be used to monitor disease progression.

The length of hospitalization or duration of IMV were not significantly different between survivors and non-survivors, but the number of days post-IMV was significantly different between non-survivors and survivors. This was because patients expired and were then removed from ventilators or patients were removed from ventilation due to a change in code status and then expired. Thus, non-survivors had a median of 0 days of post-IMV.

CXR scores correlated with a few clinical variables, most notably with LDH, RR, D-dimer, and CRP, again indicating that CXRs are informative of COVID-19 disease severity. CXR scores also correlated with the duration of hospitalization and IMV in non-survivors in the IMV group. These findings suggest that CXR scores can be used to inform hospitalization and the need for IMV. The reason we did not observe a correlation in the GF cohort is likely because there was a larger percentage of patients who were not full code in the GF cohort than in the IMV cohort, which affected the duration of the hospitalization.

### 4.4. Limitations and Future Perspectives

This study had several limitations. This was a retrospective study and thus could have residual confounding and unintentional data selection bias. Data were obtained from a single hospital, which may not generalize to other hospital settings. The decision to place patients on mechanical ventilators as well as mortality rates may depend on an individual hospital’s patient load, practice, available resources, and patients’ code status. Access to mechanical ventilators in this cohort was not a limiting factor in our hospital. Further studies using multiple institutional data and larger sample sizes as well as prospective studies are needed. Other therapies (except for mechanical ventilation) were not included and controlled for because the sample size was not adequately powered to do so. High mortality rates and severe CXR scores were associated with patients in palliative care. The therapeutic will of patients or family members could have influenced the outcomes and further studies are needed. We only correlated CXR scores with laboratory variables, hospitalization duration, IMV and post-IMV duration, but we did not correlate laboratory values with the severity of the COVID-19 disease, the length of the hospital stay, and mechanical ventilation as these have been reported extensively in the literature [28,29,30,31,32,33,34,35,36,37,38,39,40]. We did not compare different SARS-CoV-2 variants and thus this study cannot be generalized to different variants. There have been many studies demonstrating the value of utilizing machine/deep learning algorithms to detect SARS-CoV-2 infection using CXR scores [7,8,9,10,11,12,13,14,15,16,41]; it would be interesting to see whether CXR scores generated by machine/deep learning would produce similar results to those presented in this paper.

## 5. Conclusions

This study characterized the relationship between longitudinal CXR scores and clinical outcomes in patients treated on the general floors as well as patients treated with invasive mechanical ventilation. Improved CXR scores were associated with favorable outcomes. The CXR score is correlated with several clinical variables known to be associated with COVID-19 illness, hospitalization length and IMV duration. These results suggest that longitudinal CXR scores have the potential to help predict prognosis, guide treatment, monitor disease progression and allocate resources in COVID-19 circumstances.

## Figures and Tables

**Figure 1 diagnostics-13-01107-f001:**
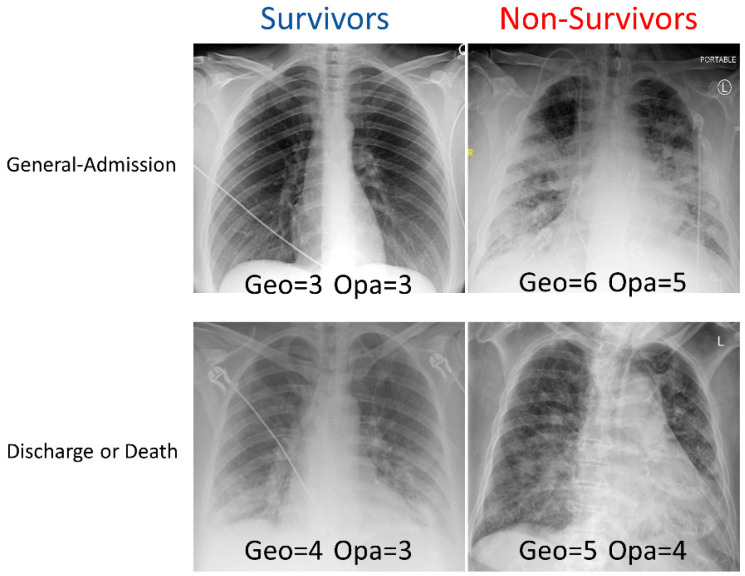
Examples of CXRs with different geographic extent and opacity scores. CXRs of COVID-19 positive patients were scored (range: 0–8) based on the extent and degree of opacities (see Methods).

**Figure 2 diagnostics-13-01107-f002:**
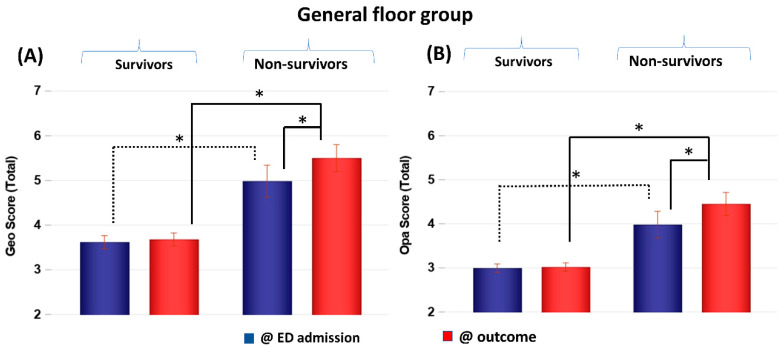
(**A**) Geographic extent (Geo) and (**B**) opacity (Opa) CXR scores of GF patients taken at emergency department admission and outcome stratified by survivors (*N* = 228) and non-survivors (*N* = 30). * *p* < 0.05. Error bars: SEM.

**Figure 3 diagnostics-13-01107-f003:**
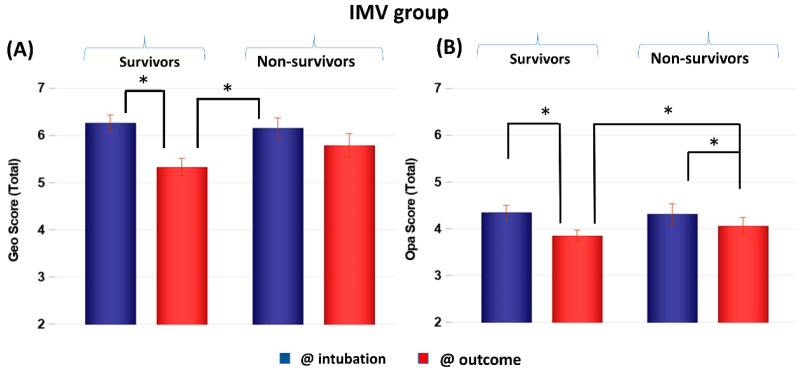
(**A**) Geographic extent (Geo) and (**B**) opacity (Opa) CXR scores of IMV patients taken at intubation and outcome stratified by survivors (*N* = 118) and non-survivors (*N* = 59). * *p* < 0.05. Error bars: SEM.

**Figure 4 diagnostics-13-01107-f004:**
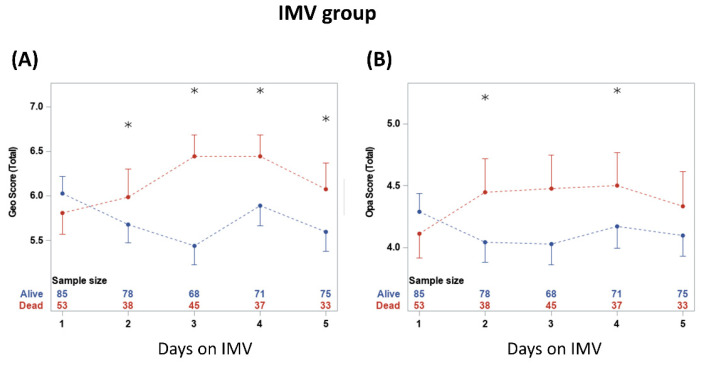
(**A**) geographic extent and (**B**) opacity chest X-ray scores taken on five consecutive days. * denotes significant difference in score value between survivors and non-survivors at a time point. The numbers show the sample sizes for each group at each time point. From two-way repeated measures ANOVA, the time and group*time had significant effects (*p* < 0.05). Significant differences from *t*-tests with the Bonferroni–Holm correction are indicated as * *p* < 0.05.

**Figure 5 diagnostics-13-01107-f005:**
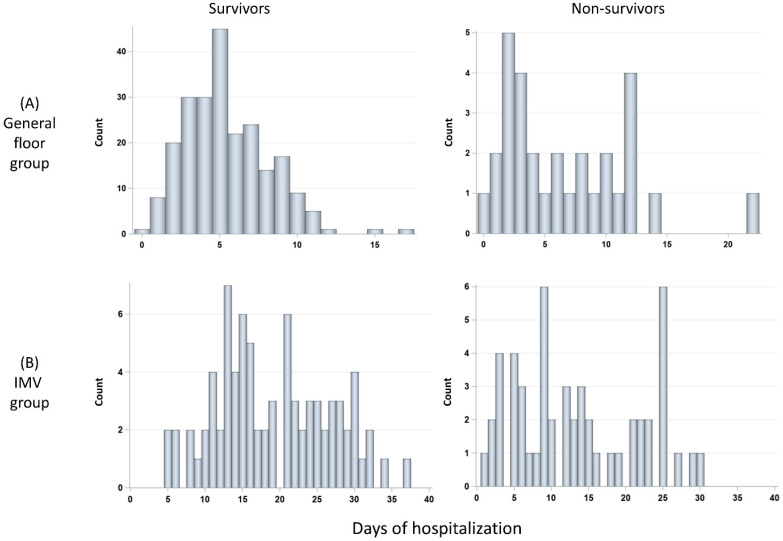
Histograms of days in the hospital for (**A**) GF group: survivors (*N* = 228) and non-survivors (*N* = 30), and (**B**) invasive mechanical ventilation (IMV) group: survivors (*N* = 85) and non-survivors (*N* = 58).

**Figure 6 diagnostics-13-01107-f006:**
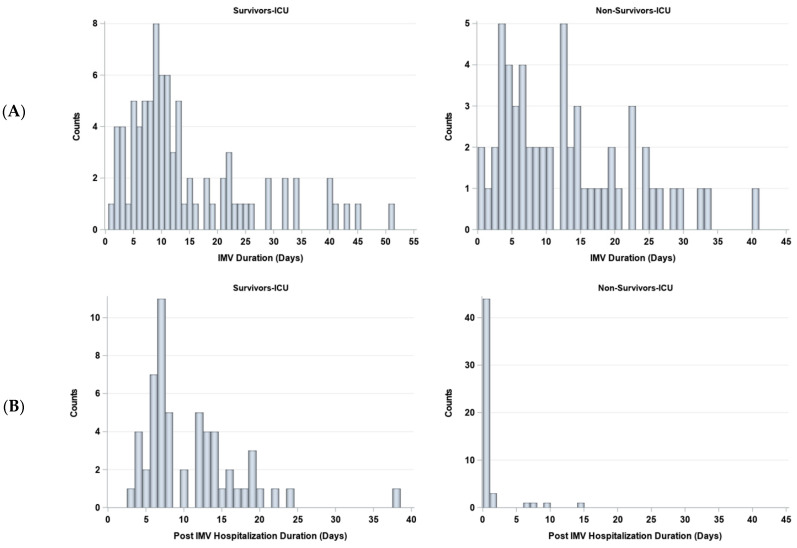
Histograms of days (**A**) on and (**B**) off invasive mechanical ventilation (IMV) for survivors (*N* = 85) and non-survivors (*N* = 58).

**Table 1 diagnostics-13-01107-t001:** Demographic characteristics and comorbidities. (A) General floor (GF) group, and (B) invasive mechanical ventilation (IMV) group, separated by survivors and non-survivors. Group comparison of categorical variables in frequencies and percentages used chi-squared or Fisher exact tests. Value inside parentheses indicate %, except for age which indicates IQR.

	GF Patients, No. (%)			IMV Patients, No. (%)		
	Survivors (*N* = 224)	Non-Survivors (*N* = 28)	*p*-Value	Survivors (*N* = 92)	Non-Survivors (*N* = 56)	*p*-Value
**Demographics**						
Age, median (IQR), y	56.0 (44.0, 69.0)	83.0 (79.0, 88.0)	<0.0001	57 (48.5, 66.0)	69 (61.5, 74.0)	<0.0001
Sex			0.9641			0.0264
Male	127 (56.7%)	16 (57.1%)		58 (63.0%)	45 (80.3%)	
Female	97(43.3%)	12 (42.8%)		34 (36.9%)	11 (19.6%)	
**Ethnicity**			0.0006			0.1617
Hispanic/Latino	72 (32.1%)	1 (3.5%)		34 (36.9%)	13 (23.2%)	
Non-Hispanic/Latino	115 (51.3%)	25 (89.2%)		42 (45.6%)	34 (60.7%)	
Unknown	37 (16.5%)	2 (7.1%)		16 (17.3%)	9 (16.0%)	
**Race**			0.0010			0.3156
Caucasian	111(49.5%)	24 (85.7%)		36 (39.1%)	25 (44.6%)	
African American	11 (4.9%)	1 (3.5%)		4 (4.3%)	2 (3.5%)	
Asian	5 (2.2%)	2 (7.1%)		6 (6.5%)	5 (88.9%)	
Others	97 (43.3%)	1 (3.5%)		46 (50.0%)	24 (42.8%)	
**Comorbidities**						
Smoking	33 (14.7%)	9 (32.1%)	0.1344	14 (15.2%)	24 (43.6%)	0.0015
Diabetes	56 (25.0%)	8 (28.57)	0.6823	25 (27.1%)	18 (32.1%)	0.5185
Hypertension	85 (37.95%)	20 (71.4%)	0.0007	36 (39.1%)	35 (62.5%)	0.0058
Asthma	15 (6.7%)	0 (0.0%)	0.1580	10 (10.8%)	2 (3.5%)	0.1147
COPD	11 (4.9%)	9 (32.1%)	<0.0001	3 (3.2%)	5 (8.9%)	0.1392
Coronary artery disease	27 (12.05%)	10 (35.7%)	0.0009	6 (6.5%)	14 (25.0%)	0.0014
Heart failure	6 (2.6%)	11 (39.2%)	<0.0001	0 (0.0%)	7 (12.5%)	0.0005
Cancer	14 (6.2%)	3 (10.7%)	0.3746	2 (2.1%)	3 (5.3%)	0.2986
Immunosuppression	15 (6.7%)	2 (7.1%)	0.9292	6 (6.5%)	2 (3.5%)	0.4414
Chronic kidney disease	17 (7.5%)	7 (25.0%)	0.0031	4 (4.3%)	7 (12.5%)	0.0667

**Table 2 diagnostics-13-01107-t002:** Vital signs and laboratory tests. (A) General floor (GF) group at ED admission and at outcome, and (B) invasive mechanical ventilation (IMV) group at intubation and at outcome, separated by survivors and non-survivors. Group comparison of continuous variables in medians and interquartile ranges (IQR) used the Mann–Whitney U test. The *p*-value column indicates significance between survivors and non-survivors. Values inside parentheses indicate IQR.

**(A)**	**GF Patients at ED Admission**			**GF Patients at Outcome**		
	**Survivors** **(*N* = 224)**	**Non-Survivors** **(*N* = 28)**	***p*-Value**	**Survivors** **(*N* = 43)**	**Non-Survivors** **(*N* = 13)**	***p*-Value**
**Vital signs, median (IQR)**						
Heart rate, bpm	88 (79, 97)	84 (74, 90)	0.1365	84 (75, 92)	80 (75, 99)	0.6572
Respiratory rate, rate/min	18 (17,21)	23 (18,28)	0.0001	18.4 (17.2, 21.5)	23 (19.3, 24.5)	0.0628
SpO2, %	95 (94, 96)	94 (90, 96)	0.0001	94 (93, 96)	93 (92, 95)	0.129
SBP, mmHg	125 (116, 139)	128 (115, 135)	0.3286	123 (116, 132)	130 (124, 136)	0.1679
Temperature, °C	37.3 (36.9, 37.7)	36.9 (36.7, 37.2)	0.0131	37.0 (36.7, 37.4)	37.1 (36.7, 37.2)	0.8652
**Laboratory findings at admission, median (IQR)**						
Alanine aminotransferase, U/L	29 (17, 50)	22 (16, 46.5)	0.3412	39 (17, 84)	20.5 (14, 47)	0.1536
Brain natriuretic peptide, ng/L	179 (31.5, 732)	1680 (673, 4726)	0.0064	198 (50, 1027)	1723 (430, 5975)	0.7402
C-reactive protein, mg/L	6.2 (22.95, 11.6)	12.8 (8.2, 20.15)	0.0001	7.3 (2.6, 16.6)	13.35 (7.2, 21.9)	0.0900
D-dimer, nmol/L	279 (179, 455)	806.5 (337, 1263)	0.0001	440 (256, 757)	947 (629, 1229)	0.0561
Ferritin, µg/L	639.95 (272, 1385)	726.9 (375, 1445)	0.8735	817 (493, 1633)	707(442, 928)	0.8140
Lactate dehydrogenase, U/L	323 (257, 393)	425 (302, 585)	0.0002	392 (276, 478)	388 (302, 540)	0.2708
Leukocytes × 10^9^/L	6.675 (5.2, 8.5)	8.3 (6.45, 10.5)	0.0179	7.6 (5, 10.1)	8.9 (5.2, 11.7)	0.3403
Lymphocytes %	14.675 (10.2, 21)	7.75 (5.7, 11.8)	0.0001	15.9 (10, 21)	7.9 (5.5, 10.6)	0.0035
Procalcitonin, ng/mL	0.1 (0.1, 0.3)	0.3 (0.2, 1.4)	0.0150	0.2 (0.1, 0.4)	0.3 (0.2, 1.7)	0.3562
Troponin, µg/L	0.04 (0.01, 0.3)	0.055 (0.01, 0.1)	0.3782	0.01 (0.001, 0.03)	0.12 (0.01, 0.1)	0.8844
**(B)**	**IMV Group at Intubation**			**IMV Group at Outcome**		
	**Survivors** **(*N* = 92)**	**Non-Survivors** **(*N* = 56)**	***p*-Value**	**Survivors** **(*N* = 92)**	**Non-Survivors** **(*N* = 56)**	***p*-Value**
**Vital signs, median (IQR)**						
Heart rate, bpm	92 (82, 100)	86 (79, 95)	0.0806	85 (72, 95)	87 (75, 103)	0.1164
Respiratory rate, rate/min	26 (23, 30)	26 (23, 30)	0.7798	22 (20, 27)	26.5 (22.3, 29.8)	0.0214
SpO2, %	94 (93, 96)	94 (92, 95)	0.0541	96 (95, 98)	94 (91, 97)	0.0001
SBP, mmHg	126 (116, 136)	125 (118, 137)	0.4545	123 (113, 135)	123 (109, 133)	0.6998
Temperature, °C	37.2 (36.9, 37.9)	37.1 (36.9, 37.5)	0.1032	37 (36.7, 37.5)	37 (36.9, 37.7)	0.1666
**Laboratory findings at admission, median (IQR)**						
Alanine aminotransferase, U/L	40 (27, 71.5)	40 (22, 59.75)	0.4866	51 (31, 89)	39.5 (23, 75.5)	0.0765
Brain natriuretic peptide, ng/L	149.5 (48, 624)	812 (156.5, 2557)	0.1794	161 (65, 624)	1063.5 (215, 3315)	0.1284
C-reactive protein, mg/L	16.5 (10.0, 26.9)	17.4 (10.1, 24.7)	0.6244	4.8 (1.2, 10.1)	11 (5.85, 25.6)	0.0001
D-dimer, nmol/L	513.5 (341, 1125)	791 (494, 2701)	0.0034	777 (427, 1430)	1284 (826, 3182)	0.0018
Ferritin, µg/L	1223 (719, 2156)	1257 (639, 2069)	0.5921	957 (692, 1624)	1330 (860, 2169)	0.1040
Lactate dehydrogenase, U/L	5488 (421, 656)	617.75 (441, 765)	0.1065	437 (334, 554)	585 (458, 744)	0.0475
Leukocytes × 10^9^/L	9.15 (6.95, 12.45)	10.82 (8.15, 14.61)	0.0317	11.1 (7.9, 14.3)	16.9 (11.3, 23.3)	0.0001
Lymphocytes %	8.05 (5.1, 12.6)	6.45 (3.75, 10.45)	0.9114	7.9 (4.4, 12.3)	3.65 (2.1, 8.5)	0.0375
Procalcitonin, ng/mL	0.3 (0.2, 0.625)	0.5 (0.25, 1.45)	0.8873	0.2 (0.1, 0.06)	0.8 (0.35, 4.35)	0.0200
Troponin, µg/L	0.01 (0.001, 0.03)	0.12 (0.01, 0.075)	0.142	0.01 (0.001, 0.04)	0.012 (0.005, 0.075)	0.0742

**Table 3 diagnostics-13-01107-t003:** Correlation (R) of geographic and opacity CXR scores with clinical variables. * *p* < 0.05. Data from admission and outcome were analyzed together.

Variable	Geographic Scores	Opacity Scores
LDH	0.41 *	0.31 *
RR	0.34 *	0.30 *
D-dimer	0.31 *	0.19 *
CRP	0.30 *	0.23 *
procalcitonin	0.27 *	0.18 *
ferritin	0.23 *	0.14 *
SpO2	−0.19 *	−0.25 *
lymphocyte	−0.19 *	−0.11 *
WBC	0.17 *	0.16
troponin	−0.04	−0.08 *
HR	−0.02	0.07 *
SBP	−0.05	−0.04
temperature	−0.05	0.000
BNP	0.03	−0.06

**Table 4 diagnostics-13-01107-t004:** Correlation (R) of geographic and opacity CXR scores for hospitalization, IMV on and post-IMV durations. * *p* < 0.05.

		Geographic Scores	Opacity Scores
Hospitalization duration	GF survivors	0.06	0.11
	GF non-survivors	−0.34	−0.27
Hospitalization duration	IMV survivors	0.33 *	0.10
	IMV non-survivors	0.49 *	0.36 *
IMV duration	IMV survivors	−0.09	−0.15
	IMV non-survivors	0.35 *	0.33 *
Post-IMV duration	IMV survivors	0.11	0.09
	IMV non-survivors	0.10	0.26

## Data Availability

Upon reasonable request via corresponding author.

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
