# Peer review of "Longitudinal Chest X-ray Scores and their Relations with Clinical Variables and Outcomes in COVID-19 Patients"

_diagnostics, 2023, doi:10.3390/diagnostics13061107_

Round 1
Reviewer 1 Report
The detailed comments on the proposed approach are enlisted in the attached document. Although the authors have written a sound and detailed approach, still, some comments are proposed below to improve the quality of the manuscript.

Reviewer 2 Report
This paper, “Longitudinal chest X-ray scores and their relations with clinical variables and outcomes in COVID-19 patients”
My comments are the following:
1.The description of the population under study and the sample with which the ANOVA analysis was made, it was prepared in detail. However, the steps of how the ANOVA analysis was performed is not clear.
2. The analysis of results and the conclusions are very appropriate in accordance with the set goals.
3. The ANOVA analysis was performed using specialized software (IBM SPSS v26). However, I consider it is necessary to include a special section as "Methodology". In where it is explained in detail that specific ANOVA was used (Randomized blocks, Latin squarer, or Factorial designs for example).
Round 2
Reviewer 1 Report
The authors improved the quality of the paper by incorporating all the reviews
and suggestions. Therefore, the paper is accepted in its current form.